# JUST ADD STRUCTURE: PROTEIN LANGUAGE MODELS COMBINED WITH STRUCTURAL EQUIVARIANCE EXCEL AT PROTEIN TASKS

**Qurat-ul-ain[1], Carlos Outeiral[1], Matteo Cagiada[1,2], Yee Whye Teh[1] & Charlotte M. Deane[1]**

[1]Department of Statistics, University of Oxford

[2]Department of Biology, University of Copenhagen

`deane@stats.ox.ac.uk`

## ABSTRACT

Accurate *in silico* prediction of protein properties, functional fitness, and mutational effects remains a central challenge in protein engineering and therapeutic design. While Protein Language Models (PLMs) successfully capture rich evolutionary and functional constraints from sequence data, they only indirectly encode the spatial and geometric information that fundamentally governs protein function. Consequently, state-of-the-art approaches typically rely on extensive fine-tuning, ensembling, or the incorporation of handcrafted structural features to achieve competitive accuracy, making them computationally expensive and difficult to scale. In this work, we demonstrate that explicit geometric modeling can substitute for, and in most cases outperform, large-scale PLM fine-tuning, with much higher parameter efficiency. Our approach, ProtEGNN, pairs PLM residue representations with a lightweight $E(3)$-Equivariant Graph Neural Network, competing with or achieving state-of-the-art performance across seven different benchmarks in protein property, mutational effect and function prediction, while needing 100–1000× fewer parameters than competing approaches. Notably, even when paired with the smallest readily available PLM, ESM2-T6 (8M parameters), ProtEGNN matches fine-tuned, sequence-only methods on mutational effect prediction, despite training orders of magnitude fewer parameters. Together, these results highlight geometric inductive bias as a powerful and scalable alternative to task-specific fine-tuning of large PLMs for protein modeling.

## 1 INTRODUCTION

Understanding how amino acid sequence gives rise to protein function and biophysical properties lies at the heart of protein engineering: the science of tailoring proteins for improved stability Tsuboyama et al. (2023), solubility Thumuluri et al. (2021), catalytic performance Yu et al. (2023) and other therapeutic traits. Recent advances in protein design Watson et al. (2023); Butcher et al. (2025); Stark et al. (2025) have expanded our ability to generate novel protein backbones and binding scaffolds, shifting the challenge towards optimising early designs into viable therapeutic and industrial products. This optimisation remains constrained by the cost and throughput of experimental protein property measurements, which are typically labour-intensive and slow. Computational approaches capable of protein analysis *in silico* are therefore essential to accelerating protein engineering and to reduce major bottlenecks in drug discovery and synthetic biology.

The most common tools used for protein analysis are Protein Language Models (PLMs) trained on large-scale sequence databases, which achieve strong performance in tasks such as property prediction, mutational-effect estimation and protein engineering (Weissenow et al., 2022; Stärk et al., 2021; Marquet et al., 2022; Lin et al., 2022). Models such as the ESM suite (ESM2, ESM3, ESMc) (Rives et al., 2019; ESM Team, 2024), ProtTrans (Elnaggar et al., 2020), Ankh3 (Alsamkary et al., 2025) and others Olsen et al. (2022); Outeiral & Deane (2024) capture evolutionary and functional constraints from protein sequences and have even been shown to implicitly reflect aspects of the ter-

tiary structure, including fold classes, residue–residue contact patterns and molecular motions (Rao et al., 2020; Vig et al., 2020; Lombard et al., 2024). However, since PLMs are trained only on amino acid sequences, this structural knowledge is indirect: it emerges from statistical regularities in sequence co-variation rather than symmetry-aware geometric reasoning. This distinction is important because many protein properties are fundamentally governed by spatial interactions (Meng et al., 2025). For example, solubility depends on surface exposure and charge distribution, and thermodynamic stability depends on core packing and hydrogen-bond networks. Numerous studies suggest that structural data (Abramson et al., 2024; Lin et al., 2022) can inform protein properties in ways that sequences alone cannot, as the three-dimensional configuration of a protein is a major determinant of its biophysical and functional characteristics (Ferreira & Castro, 2023; Vendruscolo et al., 2011).

In contrast, the predominant strategy used by state-of-the-art models for protein tasks has been to apply increasingly sophisticated fine-tuning pipelines (Jiang et al., 2024) and parameter-efficient adaptation methods on ever-larger PLMs (Schmirler et al., 2024). Furthermore, many of these approaches rely on ensembling (Thumuluri et al., 2021) or mix-and-match style frameworks that combine embeddings from multiple PLMs with diverse downstream architectures (Zhang et al., 2024; Yuan et al., 2026), often requiring thousands of experimental configurations to achieve competitive performance (Bikias et al., 2025). Moreover, extensive fine-tuning can induce catastrophic forgetting (McCloskey & Cohen, 1989), where specialization for a particular downstream task degrades the general representations learned during pretraining and leads to worse performance on other tasks compared to the original frozen model (Heinzinger et al., 2024). As the scale of PLMs has increased rapidly in recent years, growing from on the order of $10^6$ parameters to over $10^9$ parameters, these methods make task-specific adaptation of billion-parameter models neither straightforward nor computationally efficient, substantially increasing engineering complexity, training cost, and reproducibility challenges.

This emphasis on elaborate adaptation of PLMs treats sequence representations as the dominant, and often sufficient, lever for improvement, while principled geometric modeling remains underutilized. Even when structure is incorporated into PLMs, it is often done indirectly through handcrafted features, coarse distance bins, or quantazied 3D token representations that are appended to sequence embeddings rather than modeled as continuous 3D geometry (Su et al., 2023; Li et al., 2024). As a result, many methods are costly to train and adapt yet still lack explicit structural inductive biases. We challenge this fine-tune-first approach and contend that joint modeling of static PLMs representations with explicit protein structure using Equivariant Graph Neural Networks (EGNNs) Satorras et al. (2021) is a direct and efficient path to improved performance in protein tasks.

In this paper, we present ProtEGNN, a simple method to pair explicit structural information with PLMs and demonstrate that it is competitive with the leading baselines across diverse benchmarks, including many with vastly superior complexity and number of parameters. Our approach constructs graphs for each protein, where nodes correspond to 3D coordinates of carbon-alpha ($C_\alpha$) atoms of the predicted structure backbone. Each node is initialized with a feature vector given by embeddings from the final layer of a pretrained PLM; these embeddings are extracted once and treated as fixed inputs during training. Edges connect spatially proximal residues, and an EGNN updates the node features in an equivariant manner, combing multimodal signals from both sequence-derived embeddings and explicit geometry. Our contributions are as follows:

- Across 7 datasets testing protein property prediction (solubility, thermostability), mutational fitness prediction (GB1, GFP), and protein function annotation (subcellular localization), we provide a direct comparison between ProtEGNN and prior work. ProtEGNN outperforms or rivals leading methods on all evaluated tasks while using 100–1000× fewer parameters and establishes new state-of-the-art results on solubility and thermostability by substantial margins.

- We evaluate whether incorporating explicit protein structure can compensate for, or reduce the need for, adaptation of PLMs by pairing EGNNs with static residue embeddings from two pretrained PLMs of vastly different scales: the smallest readily available PLM we could find, ESM2-T6 (8M parameters) (Rives et al., 2019), and a large 6B parameter ESMc model (ESM Team, 2024). At each scale, we compare ProtEGNN to the pretrained and fine-tuned PLM, isolating the effects of model scale, fine-tuning, and explicit geometry.

- Through experiments in mutational effect prediction (GB1 and GFP), we show that modeling protein structure with an $E(3)$-equivariant graph, *even when paired with a small PLM* (ESM2-T6; 8M parameters) and a *single wild-type structure*, delivers gains that rival state-of-the-art methods relying on large-scale PLMs and task-specific fine-tuning. These results demonstrate that principled geometric inductive biases can be a more effective lever for performance than additional model scale or fine-tuning.

## 2 RELATED WORK

### 2.1 PROTEIN LARGE LANGUAGE MODELS

Many predictive methods build on pretrained PLMs by combining their embeddings with task-specific architectures. For example, PLMSol (Zhang et al., 2024) integrated embeddings from multiple PLMs using attention and recurrent modules, while LMProtein (Yuan et al., 2026) fed ESM-2 representations into a hybrid CNN, LSTM and MLP architecture. Other approaches, such as PLM-Fit Bikias et al. (2025) used combinations of ESM2 Rives et al. (2019), ProGen2 Nijkamp et al. (2023), ProteinBERT Brandes et al. (2022)) with transfer-learning methods such as feature extraction and bottleneck adapters, requiring more than 3,000 experiments. Similarly, NetSolP (Thumuluri et al., 2021) relied on extensive fine-tuning and ensembling of ESM models (Rao et al., 2020) to improve performance. While effective, these approaches often require complex pipelines and substantial computational effort. As PLMs scale, adapting them to downstream tasks has become increasingly resource-intensive, motivating parameter-efficient fine-tuning approaches. Schmirler et al. (Schmirler et al., 2024) provide detailed, "recipe"-like guidelines for parameter efficient model adaptation. Structural information has also been incorporated into protein models. Earlier work such as DeepSol (Khurana et al., 2018) used handcrafted physicochemical and structural features, whereas more recent methods embed structure directly into PLMs through discretization. For example, SaProt (Su et al., 2023) augmented the tokenizer with quantized 3D structural tokens, and ProSST (Li et al., 2024) learnt a structure-quantized latent space with sequence–structure attention. These methods inject structural cues but do so indirectly via engineered or discrete representations.

### 2.2 EQUIVARIANT GRAPH NEURAL NETWORKS

Graph Neural Networks (GNNs) incorporate structure explicitly by modeling proteins as residue- or atom-level graphs with edges defined by spatial proximity (Duval et al., 2023; Zhang et al., 2022). These models capture local and global geometric patterns and have been used for diverse tasks including protein binding site, protein-protein interaction and enzyme class prediction (Zhang et al., 2025; Hu & Ohue, 2025; Zhang et al., 2022; van der Weg et al., 2025). More recently, $E(3)$-equivariant architectures, such as Geometric Vector Perceptrons (GVPs) (Jing et al., 2020) and EGNNs (Satorras et al., 2021) have been designed to respect the inherent symmetries of data. The term 'equivariant' refers to how these networks maintain consistent behavior under transformations such as rotations, translations, and reflections. When combined with PLMs, these geometric models form multimodal pipelines in which PLM representations and structural features are fused, often via joint training of the PLM and the GNN or by augmenting graphs with handcrafted physicochemical descriptors (Wang et al., 2022; Roche et al., 2024; Tan et al., 2024).

## 3 METHODOLOGY

### 3.1 EQUIVARIANT GRAPH CONSTRUCTION

Details of how sequence and structure representations are created can be found in A.2. Following (Satorras et al., 2021), we represent each protein as a graph

$$G = (V, E, \mathbf{x}),$$

where $V = \{v_0, \ldots, v_n\}$ denotes the set of nodes corresponding to residues, $E \subseteq V \times V$ denotes the set of edges, and $\mathbf{x} = \{x_i \in \mathbb{R}^3\}_{i=0}^n$ represents the 3D coordinates of the C$\alpha$ atom associated with node $v_i$. Each layer of ProtEGNN takes as input the set of node embeddings $h^t = \{h_0^t, \ldots, h_{n-1}^t\}$, the coordinate representations $x^t = \{x_0^t, \ldots, x_{n-1}^t\}$, and the edge information $\mathcal{E} = (e_{ij})$, and

outputs updated node and coordinate representations $h^{t+1}$ and $x^{t+1}$, respectively. $t \in \{0, \dots, T-1\}$ refers to the layer number, where $T$ is a hyper-parameter. Each node $v_i$ is associated with an initial coordinate $x_i^{(0)} \in \mathbb{R}^3$, corresponding to the three-dimensional position of the $C_\alpha$ atom representing the residue in protein structure. Feature vectors for each node $v_i$ are derived from PLM embeddings and initialized as $h_i^{(0)} \in \mathbb{R}^d$. Edges are defined based on spatial proximity in 3D space using the distance matrix described above. Specifically, an edge $(v_i, v_j) \in E$ exists if the Euclidean distance between the $C_\alpha$ atoms of residues $i$ and $j$ satisfies

$$\|x_i^{(0)} - x_j^{(0)}\| < \tau.$$

where $\tau \in \{5, 10, 20\}$ Angstroms (Å) and is selected as a hyper-parameter. Equivalently, we define the neighbor set $\mathcal{N}(i) = \{j \neq i : \|x_i^{(0)} - x_j^{(0)}\| < \tau\}$ and let $E = \{(i,j) : j \in \mathcal{N}(i)\}$. We optionally allow edge attributes $e_{ij}$; in our experiments we set $e_{ij} = \varnothing$.

The layerwise equivariant message-passing is defined by the following equations:

$$m_{ij}^{(t)} = \phi_e\Big(h_i^{(t)}, h_j^{(t)}, \|x_i^{(t)} - x_j^{(t)}\|^2, e_{ij}\Big), \quad j \in \mathcal{N}(i), \tag{1}$$

$$m_i^{(t)} = \sum_{j \in \mathcal{N}(i)} m_{ij}^{(t)}, \tag{2}$$

$$x_i^{(t+1)} = x_i^{(t)} + C \sum_{j \in \mathcal{N}(i)} (x_i^{(t)} - x_j^{(t)}) \, \phi_x(m_{ij}^{(t)}), \tag{3}$$

$$h_i^{(t+1)} = \phi_h\Big(h_i^{(t)}, m_i^{(t)}\Big). \tag{4}$$

Here $\phi_e, \phi_x, \phi_h$ are edge, coordinate and node operations respectively, implemented as learnable functions using MLPs. Using relative squared distance $\|x_i^t - x_j^t\|^2$ in the edge function $\phi_e$ provides rotation and translation-invariant geometric inputs and the aggregation in Equation 2 preserves permutation invariance. Equation 3 is the crucial coordinate update step: coordinates are moved by a weighted radial vector field where each neighbor contributes the relative displacement $(x_i - x_j)$ weighted by a learned scaler $\phi_x(m_{ij})$. The constant $C$ normalizes the update to stabilize magnitudes. Equation 4 performs a permutation-equivariant update of node features using the aggregated messages $m_i^{(t)}$, allowing feature representations to be refined in each layer.

## 4 DATASETS

We evaluate ProtEGNN on seven datasets spanning three complementary protein learning regimes:

**Protein Property Prediction** Solubility (Bhandari et al., 2020) is a protein-level binary classification task, measured by ROC-AUC. To assess generalization beyond the training distribution, we additionally evaluate zero-shot solubility prediction on the Price dataset (Price et al., 2011), which shares no close homologs with PSI-Biology.

Thermostability is evaluated using two protein-level regression datasets: Meltome, from the FLIP benchmark (Dallago et al., 2021), which measures melting temperatures, $T_m$, and Stability dataset Rocklin et al. (2017), which measures protease resistance of de novo–designed mini-proteins in local mutational neighborhoods. Both tasks are evaluated using Spearman's rho (SPR).

**Mutational Effect Prediction** We evaluate on GFP from the TAPE benchmark (Rao et al., 2019) and GB1 (binding fitness) from the FLIP benchmark (Dallago et al., 2021), where each variant differs by a small number of mutations from a wild-type sequence. Both are formulated as regression tasks over variants and evaluated using SPR. In both settings, we provide only a single wild-type structure and vary only residue embeddings to reflect realistic experimental conditions.

**Protein Function Annotation** Subcellular localization is evaluated on the DeepLoc's (Almagro Armenteros et al., 2017) SetHard version (Stärk et al., 2021) as a 10-class protein-level classification task, measured by accuracy (ACC).

These benchmarks jointly probe global and local sequence–structure relationships under both sequence-diverse and mutation-centric settings. Full dataset descriptions, preprocessing steps, splits, and evaluation protocols are provided in the Appendix A.1

## 5 RESULTS

Our goal with ProtEGNN is to show that incorporating explicit 3D protein structure into pretrained PLMs is the most effective and efficient strategy for protein tasks, and should be prioritized before resorting to costly fine-tuning of large PLMs. Furthermore, to study how PLM capacity interacts with geometric modeling, we instantiated ProtEGNN with representations from two pretrained PLMs of vastly different scales: a large 6B parameter ESMc (ProtEGNN(ESMc)) and a small 8M-parameter ESM2-T6 (ProtEGNN(T6)). We trained separate models for each variant on each task. All ProtEGNN models are evaluated using 3-fold cross-validation over random seeds 97, 98, 99; except solubility which used 5-fold cross-validation following (Thumuluri et al., 2021).

### 5.1 BASELINE EXPERIMENTS

Table 1: Effect of PLM scale and explicit geometric modeling across tasks. Bold values indicate the best method.

| Sequence Model | Method | Meltome($\uparrow$) | Stability($\uparrow$) | Solubility($\uparrow$) | GFP($\uparrow$) | GB1($\uparrow$) | Sub-loc($\uparrow$) |
|---|---|---|---|---|---|---|---|
| ESM2-T6 (8M) | Pre-trained | $56.40 \pm 0.46$ | $75.00 \pm 2.02$ | $54.42 \pm 0.02$ | $63.90 \pm 0.20$ | $81.80 \pm 0.41$ | $52.00 \pm 0.61$ |
| | Fine-tuned | $58.40 \pm 0.40$ | $76.50 \pm 1.96$ | $70.31 \pm 0.04$ | $68.80 \pm 0.30$ | $88.30 \pm 0.95$ | $55.90 \pm 1.50$ |
| | ProtEGNN | $\mathbf{60.85 \pm 0.48}$ | $\mathbf{77.13 \pm 0.03}$ | $72.62 \pm 2.05$ | $\mathbf{69.48 \pm 0.12}$ | $\mathbf{89.95 \pm 0.58}$ | $\mathbf{57.21 \pm 0.62}$ |
| ESMc (6B) | Pre-trained | $58.03 \pm 0.16$ | $78.10 \pm 0.09$ | $61.00 \pm 0.03$ | $64.30 \pm 0.35$ | $83.01 \pm 0.21$ | $60.12 \pm 0.01$ |
| | Fine-tuned | $74.30 \pm 0.30$ | $\mathbf{83.80 \pm 0.36}$ | $75.90 \pm 0.05$ | $68.01 \pm 0.32$ | $89.44 \pm 0.07$ | $65.98 \pm 0.80$ |
| | ProtEGNN | $\mathbf{78.65 \pm 0.08}$ | $82.40 \pm 0.02$ | $\mathbf{78.31 \pm 2.03}$ | $\mathbf{69.13 \pm 0.02}$ | $\mathbf{89.51 \pm 0.30}$ | $\mathbf{66.81 \pm 0.09}$ |

To contextualize the performance of ProtEGNN and isolate the contribution of explicit geometry, we evaluate three settings for each PLM backbone size: a pretrained PLM used as a static feature extractor, a task-adapted PLM, and ProtEGNN, which augments fixed PLM residue embeddings with a protein structure. For the ESM2-T6 backbone, pretrained and fine-tuned results (except for solubility) are reported from (Schmirler et al., 2024). For ESMc, fine-tuning is performed using LoRA-based adaptation (Hu et al., 2022), while the pretrained baseline corresponds to frozen embeddings without task-specific updates. The second solubility dataset, Price (Price et al., 2011), is used as a holdout set to test generalization and hence, not included in this experiment.

Table 1 demonstrates that adding explicit geometric modeling consistently improves performance over sequence-only baselines, with ProtEGNN achieving the best results on five of six tasks. While increasing PLM scale benefits global property prediction, we find that on mutational tasks (GFP, GB1), the small ESM2-T6 backbone augmented with structure matches the larger ESMc. This suggests that geometric inductive biases can effectively compensate for reduced PLM capacity, particularly in local mutational contexts.

### 5.2 PROTEIN PROPERTY PREDICTION

#### 5.2.1 SOLUBILITY

Table 2a reports solubility prediction performance on the PSI-Biology dataset, together with model sizes for direct comparison. We compare ProtEGNN against Prot-T5 (Elnaggar et al., 2020), a large 1.2B parameter protein language model, ESM-MSA (Rao et al., 2021), which leverages Multiple Sequence Alignments and NetSolP (Thumuluri et al., 2021), a previous state-of-the-art solubility predictor specifically trained and tuned on the PSI-Biology dataset. NetSolP relied on both fine-tuning and ensembling of multiple ESM models Rao et al. (2020). PSI-Biology results for Prot-T5, ESM-MSA, and NetSolP are as reported in (Thumuluri et al., 2021). Despite being orders of

Table 2: Solubility prediction performance and model size comparison. Results are shown in ascending order of performance (ROC_AUC). Higher is better. Bold values indicate our method.

(a) PSI-Biology dataset (5-fold cross-validation)

| Method | # Parameters | ROC-AUC(↑) |
|---|---|---|
| Prot-T5 | 1.2B | 0.73 |
| NetSolP | 650M | 0.73 |
| **ProtEGNN(t6)** | **500K** | **0.73** |
| ESM-MSA | 100M | 0.75 |
| **ProtEGNN(ESMc)** | **1.2M** | **0.78** |

(b) Zero-shot solubility prediction on Price dataset

| Method | # Parameters | ROC-AUC(↑) |
|---|---|---|
| ProtSolM | 3.2M | 0.55 |
| PLMSoL | 7.3M | 0.60 |
| Prot-T5 | 1.2B | 0.73 |
| **ProtEGNN(t6)** | **500K** | **0.74** |
| ESM-MSA | 100M | 0.75 |
| NetSolP | 650M | 0.76 |
| **ProtEGNN(ESMc)** | **1.2M** | **0.77** |

magnitude smaller, both ProtEGNN variants perform competitively; with ProtEGNN(ESMc) setting the new state-of-the-art with a large margin while using 100x less number of parameters.

To assess out-of-distribution performance, we tested leading solubility predictors on the Price solubility dataset (Price et al., 2011), which was not used in training ProtEGNN. As shown in 2b, ProtEGNN(ESMc) achieved the best zero-shot performance with ROC_AUC of 0.77 using only 1.2M parameters, outperforming substantially larger models. Even our smaller variant, ProtEGNN(t6), remains competitive (0.74, 500K), surpassing ProtSolM (0.55, 3.2M) and PLMSoL (0.60, 7.3M). Notably, ProtSolM (Tan et al., 2024) integrates an EGNN directly with a PLM, together with handcrafted physicochemical structural descriptors, and performs end-to-end training with gradients propagated through the entire combined architecture. Similarly, PLMSol (Zhang et al., 2024) aggregated embeddings from multiple large PLMs and coupled them with an ensemble of classifiers (e.g. MLPs, Light Attention, CNN–BiLSTM), resulting in a resource-intensive approach.

### 5.2.2 STABILITY

Table 3: Thermostability prediction performance and model size comparison. Results are shown in ascending order of performance (SPR). Higher is better. Bold values indicate our method.

(a) Meltome Dataset

| Model | # Parameters | SPR(↑) |
|---|---|---|
| **ProtEGNN(t6)** | **400K** | **0.61** |
| PLM-Fit | 6.4B | 0.72 |
| Prime | 653M | 0.72 |
| SaProt | 650M | 0.72 |
| Prot-T5-FT | 3.5M | 0.72 |
| ProSST | 110M | 0.72 |
| **ProtEGNN(ESMc)** | **600K** | **0.79** |

(b) Stability Dataset

| Method | # Parameters | SPR (↑) |
|---|---|---|
| CARP | 640M | 0.72 |
| Ankh-FT | 1.2B | 0.77 |
| **ProtEGNN(t6)** | **400K** | **0.77** |
| LMProtein | Unavailable | 0.79 |
| **ProtEGNN(ESMc)** | **500K** | **0.82** |
| ESM2-T36-FT | 7.7M | 0.84 |

On the Meltome thermostability benchmark (Table 3a), we observe a strong dependence on PLM capacity: while ProtEGNN(t6) lags (0.61 SPR), scaling to ESMc achieves state-of-the-art performance (0.79 SPR). ProtEGNN outperforms resource-intensive pipelines such as PLM-Fit Bikias et al. (2025), which required over 3,000 experiments to optimize adapters across multiple PLMs and PRIME Jiang et al. (2024), which necessitated specialized pre-training on 96M bacterial sequences. ProtEGNN also surpasses Prot-T5-FT (Schmirler et al., 2024), which fine-tuned 3.5M parameters of the 1.2B Prot-T5 (Elnaggar et al., 2020) using LoRA (Hu et al., 2022). Crucially, ProtEGNN(ESMc) also surpasses structure-aware baselines such as ProSST Li et al. (2024), which learnt disentangled sequence-structure attention, and SaProt (Su et al., 2023), which fused discrete structural tokens

into PLM training. In contrast, ProtEGNN achieves top performance by training only a lightweight EGNN (≈600K parameters) on a frozen PLM backbone, demonstrating that explicit geometric modeling is a far more efficient alternative to extensive architectural search or specialized pre-training.

On the second stability benchmark (Table 3b), ProtEGNN(t6) achieves an SPR of 0.77, matching the performance of the 1.2B-parameter finetuned Ankh-FT despite using only 400K trainable parameters. Scaling to ProtEGNN(ESMc) further improves performance to 0.82 SPR, surpassing CARP (0.72) (Yang et al., 2024) which combined PLMs with Convolutional architecture, and LM-Protein (0.79) (Yuan et al., 2026), a hybrid architecture that stacks CNNs, LSTMs, and MLPs on top of ESM-2 embeddings. The LoRA finetuned ESM2-T36-FT (Schmirler et al., 2024) achieves the highest score (0.84), albeit using orders-of-magnitude more parameters.

## 5.3 MUTATIONAL EFFECT PREDICTION

Table 4: GFP and GB1 mutational effect prediction performance and model size comparison. Results are shown in ascending order of performance (SPR). Higher is better. Bold values indicate our method.

<table>
<tr><td colspan="3" align="center">(a) GFP</td><td colspan="3" align="center">(b) GB1</td></tr>
<tr><td>Model</td><td># Parameters</td><td>SPR (↑)</td><td>Model</td><td># Parameters</td><td>SPR(↑)</td></tr>
<tr><td>Prot-T5</td><td>1.2B</td><td>0.61</td><td>SaProt</td><td>650M</td><td>0.81</td></tr>
<tr><td>Ankh3</td><td>5.7B</td><td>0.65</td><td>PRIME</td><td>653M</td><td>0.82</td></tr>
<tr><td>LM-GVP</td><td>Unavailable</td><td>0.68</td><td>ESM2-T48</td><td>15B</td><td>0.85</td></tr>
<tr><td>ProtEGNN(ESMc)</td><td>650K</td><td>0.69</td><td>PLM-Fit</td><td>6.4B</td><td>0.88</td></tr>
<tr><td>Ankh-FT</td><td>4.9M</td><td>0.70</td><td>ESM2-T36-FT</td><td>7.7M</td><td>0.89</td></tr>
<tr><td>ProtEGNN(t6)</td><td>500K</td><td>0.70</td><td>ProtEGNN(ESMc)</td><td>1M</td><td>0.90</td></tr>
<tr><td></td><td></td><td></td><td>ProtEGNN(t6)</td><td>400K</td><td>0.90</td></tr>
<tr><td></td><td></td><td></td><td>Ankh3</td><td>5.7B</td><td>0.90</td></tr>
</table>

For each dataset, we use a *single wild-type structure* to represent all variants, fixing the underlying geometry across the mutational landscape. Even without modeling the explicit mutation-induced structural rearrangements, our equivariant geometric model captures local physical constraints that remain informative under sequence perturbations, enabling state-of-the-art mutational effect prediction.

### 5.3.1 GFP

Table 4a reports results on the GFP fluorescence landscape. ProtEGNN(T6) achieves state-of-the-art performance (SPR = 0.70), tying with Ankh-FT model (Schmirler et al., 2024), while training only 500K task-specific parameters on top of a small 8M-parameter PLM backbone. Notably, Ankh-FT applied LoRA on top of the 1.9B parameter Ankh backbone, resulting in millions of trainable parameters. ProtEGNN achieves the same performance using fewer parameters (4.9M vs 500K) and much smaller PLM backbone (1.9B vs 8M).

Both ProtEGNN variants outperform substantially larger models such as Prot-T5 (1.2B parameters) (Elnaggar et al., 2020) and Ankh3 (5.7B parameters) (Alsamkary et al., 2025). ProtEGNN also performs better than LM-GVP (Wang et al., 2022) (0.68), which stacked an EGNN in front of a PLM (Elnaggar et al., 2020) and jointly trained the models, modifying and fine-tuning the PLM embeddings.

### 5.3.2 GB1

We observe a similar pattern in the GB1 binding landscape (Table 4b). Both ProtEGNN(t6) and ProtEGNN(ESMc) reach an SPR of 0.90, matching the best-performing 5.7B parameter Ankh3 model (Alsamkary et al., 2025). This parity is achieved with as few as 400K trainable parameters in

ProtEGNN(t6). Notably, ESM2-T36-FT, as reported in (Schmirler et al., 2024) applies LoRA to a 3B parameter PLM backbone, reducing the number of trainable parameters to only 7.7M; however, despite this adaptation, it still underperforms ProtEGNN, highlighting the limitations of parameter-efficient fine-tuning when structural inductive biases are absent. Crucially, our results show that adding a single wildtype structure to even the smallest 8M parameter backbone delivers predictive power equivalent to multi-billion parameter sequence models without the need for extensive fine-tuning or structure predictions for every variant.

## 5.4 PROTEIN FUNCTION ANNOTATION

### 5.4.1 SUBCELLULAR LOCALIZATION

Table 5: Subcellular Localization performance and model size comparison. Results are shown in ascending order of performance (ACC). Higher is better. Bold values indicate our method.

| Subcellular Localization | | |
| --- | --- | --- |
| Model | # Parameters | ACC($\uparrow$) |
| Prost-T5 | 2.8B | 0.57 |
| **ProtEGNN(t6)** | **400K** | **0.57** |
| Prot-T5 | 3B | 0.65 |
| LA-Prot-T5 | 3B | 0.65 |
| **ProtEGNN(ESMc)** | **500K** | **0.67** |
| ESM2-T36 | 7.7M | 0.68 |

On the subcellular localization benchmark, Table 5, ProtEGNN achieves competitive accuracy with orders-of-magnitude fewer parameters than large PLM baselines. In particular, ProtEGNN matches the performance of Prosst-T5 (Heinzinger et al., 2024) (ACC = 0.57) while using only 400K parameters. Notably, even though Prosst, a fine-tuned derivative of ProtT5 (Elnaggar et al., 2020), incorporates structural information, its reduced performance relative to ProtT5 and LA-Prot-T5 (Stärk et al., 2021) (0.57 vs. 0.65) highlights the effect of task specialization through fine-tuning, where adapting a PLM for one objective can degrade its performance on others.

In contrast, ProtEGNN(ESMc) attains an accuracy of 0.67 with just 500K parameters, closely approaching the performance of the fine-tuned ESM2-T36 (0.68) reported in (Schmirler et al., 2024). It is important to note that (Schmirler et al., 2024) employed parameter efficient fine tuning techniques to reduce the trainable parameters of ESM2-T36 from 3B to 7.7M. However, even parameter-efficient fine-tuning techniques approaches incur significantly higher parameters compared to our method, which simply adds a lightweight geometric head to frozen PLM representations. These results reinforce the idea that adding explicit structural inductive biases is a more robust and efficient alternative to fine-tuning large PLMs, even when parameter-efficient techniques are used.

## 6 DISCUSSION

Our experiments convey a clear and actionable message: explicit, equivariant geometric modeling can substitute for, and in most cases outperform, large-scale PLM fine-tuning, while being 100-1000$\times$ more parameter-efficient. Our findings expose a broader limitation of current practice: fine-tuning large PLMs for each downstream task is not a viable long-term strategy. Beyond the computational burden, extensive fine-tuning sometimes leads to task specialization at the cost of generality, as seen in the case of ProSST where subcellular localization performance degrades after fine-tuning. Our results on mutational landscapes demonstrate that geometric deep learning offers distinct advantages for modeling proteins, enabling lightweight models to capture local physical constraints and deliver much of the predictive benefit usually attributed to massive, fine-tuned PLMs

We emphasize that our efficiency claims refer specifically to the number of trainable parameters and task-specific optimization cost. Structure prediction is performed once per protein as an offline pre-

processing step and can be amortized across tasks, in contrast to repeated fine-tuning of large PLMs for each downstream objective. Moreover, as our pipeline relies on predicted structures, future work should explore uncertainty-aware inference to mitigate sensitivity to structural quality; especially for disordered or membrane proteins which are poorly predicted by current structure prediction methods. We also encourage exploration into multi-task EGNN heads and interpretability methods that connect EGNN message passing to concrete biophysical mechanisms. Broadly, our findings underscore the critical importance of appropriate inductive biases in model design. We contend that prioritizing higher-fidelity representations and biology-aware architectures will yield significantly greater dividends in protein modeling than indiscriminate parameter scaling or elaborate adaptation of PLMs.

## MEANINGFULNESS STATEMENT

Proteins are the molecular machines of life, with their function defined by the intricate interplay between amino acid sequence and 3D structure. While Protein Language Models have successfully leveraged vast sequence databases to capture evolutionary context, they often lack explicit physical inductive bias which no amount of task specific adaption can achieve. With the growing availability of structural data, we argue that the path forward lies in multimodal synergy between these two biophysical representations. Our work demonstrates that anchoring sequence embeddings in explicit geometry captures critical environmental contexts missed by massive sequence-only models. This proves that meaningful representations of life emerge from the combination of evolutionary history and physical structural constraints.

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

# A APPENDIX

## A.1 DATASETS

We evaluate ProtEGNN on seven datasets spanning three protein task regimes: protein property prediction, mutational-effect prediction, and protein function annotation. Together, these tasks probe global and local sequence–structure relationships under both sequence-diverse and mutation-centric settings making them a comprehensive testbed for assessing the value of explicit 3D geometry.

### A.1.1 PROTEIN PROPERTY PREDICTION

To assess the prediction of intrinsic protein properties that depend on global structure and physico-chemical context, we evaluate solubility and stability.

**Solubility** We use the PSI Biology solubility dataset (Bhandari et al., 2020), curated and cleaned by (Thumuluri et al., 2021), containing 11,226 *E. coli*–expressed proteins labeled as soluble or insoluble. We follow the standard five-fold, label-balanced cross-validation protocol with sequence identity capped at 25%. To evaluate out-of-distribution generalization, we additionally test on the independent Price dataset (Price et al., 2011) which contains 1323 highly expressed proteins. This dataset also does not share any sequences with identity greater than 25% to the PSI Biology dataset, ensured using USEARCH (Edgar, 2010). Solubility is a graph-level binary classification task with ROC_AUC as the metric.

**Stability** We evaluate protein stability using two complementary datasets that probe distinct notions of thermostability. The first is the Meltome Atlas dataset (referred to as Meltome in results), which

contains $\sim 23{,}300$ protein sequences with melting temperatures ($T_m$) measured via mass spectrometry. $T_m$ provides a continuous-valued proxy for intrinsic thermostability and reflects both local packing interactions and long-range structural organization. We follow the train-test splits from the FLIP benchmark (Dallago et al., 2021), which enforces redundancy reduction by clustering sequences at 20% pairwise sequence identity. Predictions on Meltome are formulated as a graph-level regression task and evaluated using SPR.

In contrast, the dataset introduced by Rocklin et al. (2017) (referred to as Stability in results) from the TAPE (Rao et al., 2019) benchmark, containing $\sim 69{,}000$ records, measures stability indirectly through protease resistance of de novo–designed mini-proteins, capturing relative fitness within local mutational neighborhoods rather than absolute thermodynamic stability. We adopt the standard TAPE split, where training and validation sets are drawn from four rounds of experimental design, and the test set consists of single-point mutants (Hamming distance 1) around 17 selected high-performing designs. This task is formulated as a graph-level regression problem analogous to learning mutational fitness landscapes evaluated using SPR.

### A.1.2 MUTATIONAL EFFECT PREDICTION

Fine-tuned PLMs have been shown to perform well on mutational effect prediction, as they implicitly capture evolutionary constraints and co-variation patterns that correlate strongly with fitness changes (Glaser & Brägelmann, 2025). Given this strong baseline, our aim was to test whether explicit geometric modeling with EGNNs can be competitive in this regime. In our setup, we provide the ProtEGNN with a single wild-type structure and vary only the residue embeddings for each mutant sequence, keeping the geometry fixed. This reflects a realistic experimental scenario, where typically only one high-quality structure is available and variant-specific structures are either unavailable or indistinguishable due to the limited sensitivity of structure predictors to single or few mutations.

**GFP** From the TAPE benchmark (Rao et al., 2019), we use the fluorescence dataset (GFP), a regression task where $\sim 54{,}000$ sequences are mapped to their log-fluorescence intensity. Training variants are within Hamming distance 3 of the wild type, while test variants contain four or more mutations, explicitly probing generalization to unseen mutation combinations. GFP is a node-level regression task measured by SPR.

**GB1** We evaluate on the GB1 landscape from the FLIP benchmark (Dallago et al., 2021), which measures binding affinity of $\sim 8{,}700$ variants of the immunoglobulin-binding domain of Protein G. Mutations are introduced at four positions, producing a highly epistatic landscape. We use the standard 3-vs-rest split, where single, double, and triple mutants are used for training and more distant variants for testing. GB1 is a node-level regression task measured by SPR.

### A.1.3 PROTEIN FUNCTION ANNOTATION

**Subcellular Localization** To evaluate functional annotation, we use the harder version of DeepLoc dataset (Almagro Armenteros et al., 2017), SetHard, developed by (Stärk et al., 2021). This dataset, $\sim 11{,}700$ records, frames subcellular localization as a 10-class per-protein classification task. Sequences are filtered to remove redundancy above 20% identity between splits using MMseqs2, ensuring that predictions require generalization beyond close homologs. Subcellular localization (referred to as Sub-loc in results) is a graph-level multi-classification task measured by Accuracy (ACC).

## A.2 SEQUENCE AND STRUCTURE REPRESENTATIONS

For sequence information, we extract last-layer, per-residue embeddings from two pretrained PLMs of different scales: ESM2-T6 (8M parameters) and ESMc (6B parameters). For a protein of length $n$, these embeddings form a matrix in $\mathbb{R}^{n \times d}$, where the embedding dimension $d$ is 320 for ESM2-T6 and 2560 for ESMc. During training, gradients from the downstream loss are not propagated back to the PLM; the sequence representations are extracted once and used as fixed node features.

Structural information is obtained by predicting protein 3D coordinates using ESMFold (Lin et al., 2022). Each residue is represented by its C$\alpha$ atom, which provides a consistent and compact representation of the protein backbone. Using the C$\alpha$ coordinates, we compute an $n \times n$ pairwise distance matrix, where each entry $d_{ij}$ denotes the Euclidean distance between residues $i$ and $j$.

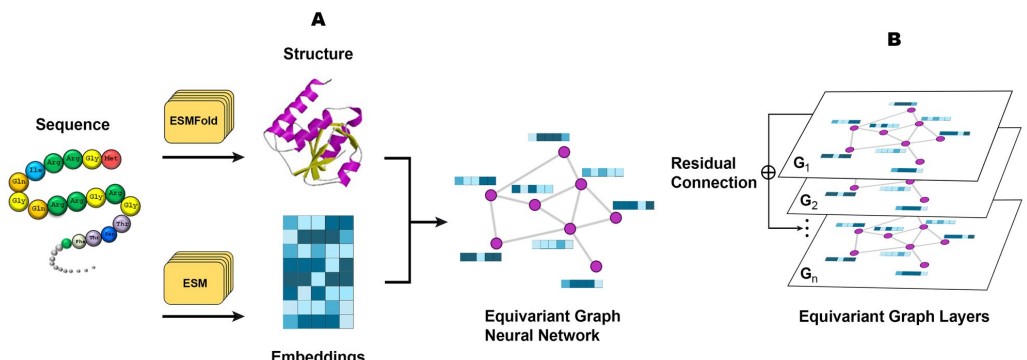

Figure 1: Integration of protein sequence and structure data using a EGNN architecture in ProtEGNN. In **Panel A**, a protein sequence is processed through two distinct pathways: ESMFold predicts the 3D structure of the protein, while ESM (ESMc or ESM2-T6) generates sequence embeddings. The predicted structure and sequence embeddings are then combined to construct a graph representation, where nodes correspond to amino acids and edges represent spatial relationships. In **Panel B**, the graph is passed through multiple EGNN layers. Residual connections are used between layers to maintain information flow and improve learning stability.

## A.3 HYPER-PARAMETERS

Table 6: Hyperparameters in `ProtEGNN`.

| Parameter | Default | Possible values | Explanation / where used |
|---|---|---|---|
| `batch_size` | 2 | int{2,4,8,16} | PyG DataLoader batch size. |
| `epochs` | 5 | int{5,10,20,30,40,50,60} | Training epochs for cross validated runs. |
| `learning_rate` | 1e-5 | float(log-uniform1e-4..1e-2) | AdamW learning rate. |
| `weight_decay` | 1e-3 | float({1e-6,1e-5,3e-5,1e-4}) | AdamW weight decay. |
| `scheduler` | CosineAnnealingWarmRestarts | {CosineAnnealingLR,CosineAnnealingWarmRestarts} | LR scheduler choice (eta_min is set to LR/100 in both). |
| `val_maximize` | SPR | ACC, ROC_AUC, SPR | Early-stopping/model-selection metric. |
| `early_stopping_delta` | 0.01 | float$\geq$ 0 | Minimum improvement threshold for early stopping. |
| `angstroms` | 10 | float{5,10,20}) | Distance cutoff (Å) for creating edges |
| `model_layers` | 3 | int({2,3,4,5,6,8}) | Number of EGNN message-passing layers. |
| `activation` | silu | {silu,relu} | Nonlinearity in EGNN and heads. |
| `egnn_hidden_nf` | 128 | int({64,96,128,142,256,320}) | Hidden width inside EGNN layers. |
| `egnn_output_dim` | 96 | int({64,96,128,256}) | Output node embedding dim from EGNN backbone. |
| `egnn_norm` | None | None or layernorm | If `layernorm`, inserts LayerNorms in EGNN MLPs. |
| `use_adaptive_pooling` | True | {True,False} | If True, learns weights over mean/max/sum pooling (graph tasks). |
| `pool_attention_hidden` | [64] | int({32,64,128,256}) | Hidden dim of pooling-attention MLP. |
| `fc_head_hidden_dims` | [256,128,64] | list[int] | Hidden layer sizes for graph-level head (`MLPHead`). |

| Parameter | Default | Possible values | Explanation / where used |
|---|---|---|---|
| `node_head_hidden_dims` | [64] | list[int] | Hidden layer sizes for node-level head (`MLPHead`). |
| `head_dropout` | 0.15 | floatin[0,1](0.1--0.3) | Dropout in graph-level head. |
| `node_dropout` | 0.3 | floatin[0,1](0.2--0.4) | Dropout in node-level head. |
| `pos_weight` | 0.67 | {0.5,0.67,0.8}) | Positive-class weight for BCEWithLogits (binary graph tasks). |
| `use_focal_loss.enabled` | False | {True,False} | Switch focal loss for binary classification. |
| `use_focal_loss.focal_alpha` | 0.25 | {0.25,0.5} | Focal loss $\alpha$ (binary). |
| `use_focal_loss.focal_gamma` | 2.0 | {1.0,2.0} | Focal loss $\gamma$ (binary). |
| `use_huber_loss.enabled` | False | {True,False} | Switch Huber loss for regression. |
| `use_huber_loss.huber_delta` | 1.0 | {0.5,1.0,1.5} | Huber $\delta$ threshold. |
| `use_label_smoothing.enabled` | True | {True,False} | Enables label smoothing loss for multiclass. |
| `use_label_smoothing.label_smoothing_epsilon` | 0.1 | {0.05,0.1,0.15} | Label smoothing strength $\epsilon$. |
| `use_focal_loss_multiclass.enabled` | False | {True,False} | Switch focal loss for multiclass. |
| `use_focal_loss_multiclass.gamma` | 2.0 | {1.5,2.0,2.5,3.0} | Multiclass focal $\gamma$. |

