# OpenReview forum: "JUST ADD STRUCTURE: PROTEIN LANGUAGE MODELS COMBINED WITH STRUCTURAL EQUIVARIANCE EXCEL AT PROTEIN TASKS"
_ICLR.cc/2026/Workshop/LMRL — ICLR 2026 Workshop LMRL Poster_

### Official Review · Reviewer_YNQk · 2026-02-21
**The authors propose ProtEGNN where they combine structural information with pLMs by constructing graphs. They evaluate the approach across 7 tasks and achieve competitive performance compared to larger sequence only protein models such as ProtT5.**

**Rating:** 4
**Confidence:** 4

**Review:**

The motivation behind adding explicit structural information along with sequence embeddings from pre-trained pLMs towards predicting different protein properties is clear since quantization may compress information and is a noisy approximation of the structure features (e.g. SaProt, ESM3). The proposed method of using C-alpha coordinates, embeddings from different ESM models and constructing graph is simple and elegant. The method section is easy to follow. The results ( Table 1) show that the structure information on a small model is already good but can be added to larger pLMs to improve results. The advantage of the method is no need to construct equi-variant GNNs or explicit atom level graphs and it is a plug and play application.

Below are some comments
1. The approach relies on predicted structures from ESMfold which means the model is biased to this upstream model. How robust is the model to structures from different prediction models such as AF3 or Boltz? A comparison would reveal sensitivity to structural errors if any.
2. The approach claims that tokenization of structure is not as effective but no comparison was done against SaProt, ProstT5 or more recent multimodal models like ESM3 where there is an explicit geometric attention module in addition to structure tokenization. Even if the proposed approach cant beat these, to know the gap would be helpful. If it beats then the claim on the paper would be stronger.
3. There are other structure aware models such as GearBind which model interface and can be combined with residue embeddings. No comparison was done to show which way of structure pairing was most effective.

If the authors could address the comments it would make the paper much stronger.

---

### Official Review · Reviewer_BmEU · 2026-02-22
**Review of Submission 74**

**Rating:** 7
**Confidence:** 4

**Review:**

The paper introduces ProtEGNN to integrate 3D structural information with evolutionary data captured by PLMs like ESMc and ESM2-T6. The method represents proteins as graphs of C_{\alpha} atoms, where nodes are initialized with fixed sequence embeddings and updated through equivariant message-passing layers. Results show that ProtEGNN sets new performance records in solubility and thermostability, often by substantial margins. Furthermore, the model shows that a small 8M-parameter backbone augmented with structure can match or exceed the performance of fine-tuned billion-parameter models in mutational fitness tasks.

The framework achieves state-of-the-art performance across multiple protein tasks while requiring 100–1000× fewer trainable parameters than leading baselines. It successfully demonstrates that explicit 3D geometric inductive biases can compensate for smaller model scales, allowing an 8M-parameter model to rival 6B-parameter counterparts.

However, the model's reliance on predicted structures from ESMFold means its accuracy is potentially limited by the quality of these predictions, particularly for disordered or membrane proteins. Additionally, the workflow necessitates a separate, offline structure prediction step for every protein, which introduces a preprocessing bottleneck that sequence-only models do not face.

---

### Official Review · Reviewer_SJLW · 2026-02-25
**Combining protein language models with E(3)-equivariant structural modeling yields parameter-efficient, state-of-the-art performance across diverse protein prediction tasks**

**Rating:** 7
**Confidence:** 3

**Review:**

This work presents a clear, well-motivated argument that protein language models combined with structural equivariance excel at protein tasks. The paper is technically high level, experimentally thorough, and conceptually coherent, demonstrating that pairing frozen PLM embeddings (e.g. ESM2 and ESM variants) with an E(3)-equivariant graph neural network yields performance that rivals or surpasses large-scale fine-tuned models across diverse benchmarks. The originality lies not in introducing a new PLM or a novel equivariant architecture but in systematically challenging the fine-tune-first methodology and showing that explicit geometric inductive bias can substitute for extensive parameter adaptation with 100–1000× fewer trainable parameters. The significance include the work provides a scalable and parameter-efficient alternative to costly fine-tuning pipelines, with consistent gains in solubility, thermostability, mutational effect prediction, and function annotation. Key strengths include strong empirical benchmarking across seven datasets, careful comparison of model scale versus geometric modeling, parameter-efficiency analysis, and a compelling demonstration that even small PLMs augmented with equivariant structure can match multi-billion-parameter baselines. Limitations include reliance on predicted structures (with potential sensitivity to structural quality), limited exploration of structural uncertainty or disordered proteins, absence of multi-task evaluation within a unified model, and the fact that structure prediction cost—though amortized—is still required. Overall, the work convincingly argues that incorporating symmetry-aware geometric modeling alongside PLMs is a more principled and scalable direction than increasingly elaborate fine-tuning of large sequence-only models.

---

### Meta-Review · Area_Chair_XNbQ · 2026-02-28

**Recommendation:** Accept (Poster)
**Confidence:** 4

**Metareview:**

Accept

---

### Decision · Program_Chairs · 2026-03-02

**Decision:**

Accept (Poster)

**Comment:**

Please see the meta-review.